# TranSlowDown: Efficiency Attacks on Neural Machine Translation Systems

## Abstract

Neural machine translation (NMT) systems have received massive attention from academia and industry. Despite a rich set of work focusing on improving NMT systems' accuracy, the less explored topic of *efficiency* is also important to NMT systems because of the real-time demand of translation applications. In this paper, we observe an inherent property of NMT system, that is, NMT systems' efficiency is related to the output length instead of the input length. Such property results in a new attack surface of NMT system—an adversary can slightly changing inputs to incur significant amount of redundant computations in NMT systems. Such abuse of NMT systems' computational resources is analogous to the *denial-of-service* attacks. Abuse of NMT systems' computing resources will affect the service quality (*e.g.,* prolong response to users' translation requests) and even make the translation service unavailable (*e.g.,* running out resource such as batteries of mobile devices). To further the understanding on such efficiency-oriented threats and raise community's concern on the efficiency robustness of NMT systems, we propose a new attack approach, `TranSlowDown`, to test the efficiency robustness of NMT systems. To demonstrate the effectiveness of `TranSlowDown`, we conduct a systematic evaluation on three public-available NMT systems: Google T5, Facebook Fairseq, and Helsinki-NLP translators. The experimental results show that `TranSlowDown` can increase NMT systems' response latency up to 1232% and 1056% on Intel CPU and Nvidia GPU respectively by inserting only three characters into existing input sentences. Our results also show that the adversarial examples generated by `TranSlowDown` can consume more than *30* times battery power than the original benign example. Such results suggest that further research is required for protecting NMT systems against efficiency-oriented threats.

## 1 Introduction

Recently, Neural Machine Translation (NMT) systems have received massive attention from academia and industry (Bahdanau et al., 2015; Kalchbrenner et al., 2016; Vaswani et al., 2017; Belinkov & Bisk, 2018). NMT systems overcome many weaknesses of traditional phrase-based translation models and can capture long dependencies in sentences; thus, they are widely used in commercial translation systems. For example, Microsoft has deployed the NMT systems in many commercial products since 2016 (Hassan Awadalla et al., 2017; 2018a;b; Gu et al., 2018); Google Translate claims to have translated over 100 billion words daily in 109 languages (Turovsky, 2016; Caswell & Liang, 2020; Pitman, 2021).

For NMT systems, efficiency is critical because of translation applications' real-time demand (Tu et al., 2017; Guo et al., 2019; Xia et al., 2019). However, it is unknown whether NMT systems are robust against efficiency-oriented adversarial pressure. Despite a rich set of works (Cheng et al., 2020; Jin et al., 2020; Belinkov & Bisk, 2017; Cheng et al., 2019; Wu et al., 2018; Yang et al., 2017) evaluate NMT systems' accuracy robustness through maximizing the errors, understanding the NMT systems' efficiency robustness has not received much attention.

In order to study NMT systems' efficiency robustness, we first need to figure out what factors will affect NMT's efficiency. In this paper, we observe a natural property of NMT systems, *i.e.,*, NMT's computational consumption is volatile for different inputs because NMT systems invoke the underlying decoder with non-deterministic numbers of iterations to generate output tokens (Vaswani et al.,

2017; Liu et al., 2020). This property exposes a new vulnerability of NMT systems, an adversary can design specific inputs to cause enormous computation overhead in NMT systems, thus wasting the computational resources of NMT systems. Based on such observation, in this paper, we investigate: *Can adversary make slight modification on textual inputs to increase NMT systems' computational consumption and decrease NMT systems' efficiency? If so, how severe the efficiency degradation can be?*

**New Vulnerability.** We consider a new attack surface of NMT models, analogous to the vulnerabilities leading to the denial of service (DoS) attacks (Zhang et al., 2015; Qin et al., 2018) that have plagued the security committee for decades. Specifically, the adversary's goal is to decrease the victim NMT system's efficiency (*i.e.,* response latency, energy consumption) with unnoticeable perturbations on the inputs to the system. This attack will result in devastating consequences for many real-world applications. For example, abusing computational resources on commercial machine translation service providers (*e.g., Huggingface* (Wolf et al., 2020)). An efficiency attack will cause enormous redundant computational resources and affect the service quality of benign users. Furthermore, abusing computational resources on mobile devices or IoT devices might shorten the battery charge cycle and result in the unavailability of the devices.

**New Attack.** In this paper, we evaluate NMT systems' efficiency robustness by generating adversarial inputs that consume much greater amount of computation resources than normal inputs on NMT systems. Specifically, we propose `TranSlowDown`, which is based on the observation that the source of vulnerability is that NMT systems' efficiency is related to the output length instead of the input length. Specifically, NMT systems iteratively compute the output token until the systems generate the particular end of sentence (EOS) token. Thus, we design a novel algorithm to search for a minimal perturbation to delay the appearance of EOS. Specifically, `TranSlowDown` can generate both *token-level* and *character-level* perturbations. After applying the minimal perturbation on the benign sentences, the probability of EOS of output token will decrease, resulting longer output that costs NMT systems more computational resources.

**Evaluation.** To evaluate the effectiveness of `TranSlowDown`, we use `TranSlowDown` to attack three public-available NMT systems (*i.e.,* Google T5 (Raffel et al., 2019), Facebook FairSeq (Liu et al., 2020), and Helsinki-NLP), and compare `TranSlowDown` against two accuracy-based attack algorithms. We first measure the floating-point operations (FLOPs) and response latency of the victim NMT systems when translating benign and adversarial examples. Then, we apply I-FLOPs and I-Latency (defined in Equation 4) to quantify the severity of efficiency degradation caused by generated adversarial examples. The evaluation results show: for the token-level attack, `TranSlowDown` generate adversarial examples increase victim NMT systems' FLOPs, CPU latency, GPU latency up to 2131.92%, 2403.21%, and 2054.83% respectively with only perturbing two tokens in sentences; for the character-level attack, `TranSlowDown` generated adversarial examples increase victim NMT systems' FLOPs, CPU latency, GPU latency up to 1102.86%, 1232.71%, and 1056.88% respectively with only inserting three characters in sentences. We also form a real-world case study to demonstrate the negative impact of the efficiency threat. Specifically, we deploy Google's T5 on a mobile device and investigate how adversarial examples affect the mobile device's battery power consumption. The results show that by inserting only one character into the benign sentence, the adversarial examples generated by `TranSlowDown` can consume more than *30* times battery power than the original benign example. Our results suggest that further research is required for protecting NMT systems against this emerging security threat.

## 2 BACKGROUND & RELATED WORK

### 2.1 NEURAL MACHINE TRANSLATION SYSTEMS

Neural machine translation (NMT) (Vaswani et al., 2017; Liu et al., 2020) systems apply neural networks to approximate the conditional probability $P(Y|X)$, where $X = [x_1, x_2, \cdots, x_m]$ is the source token sequences and $Y = [y_1, y_2, \cdots, y_n]$ is the target token sequences. As shown in Figure 1, a typical NMT system consists of an encoder $f_{en}(\cdot)$ and a decoder $f_{de}(\cdot)$. The encoder encodes the source input $X$ into hidden representation $H$, then the decoder starts from a special token (SOS), and iteratively accesses $H$ for an auto-regressive generation of each token $y_i$ until the end of sequence token (EOS) is reached. An important observation from the NMT working mechanisms is that NMT will iteratively running the decoder $f_{de}$ to generate a output token until EOS is

reached. Thus, if the generated output sequence is longer, the NMT will consume more computational resources and becomes less efficient. In our example, the NMT system run the decoder four times to generate the output.

## 2.2 DNN's Efficiency

Recently, the efficiency of deep neural networks (DNNs) has raised a huge concern because of their substantial *inference-time* computational costs. To reduce DNN' *inference-time* computational costs and make it feasible to applying DNNs for real-time applications, many existing work has been proposed. There are two main techniques: The first type (Howard et al. (2017); Zhang et al. (2018)) of the techniques prune the DNNs offline to identify important neurons and

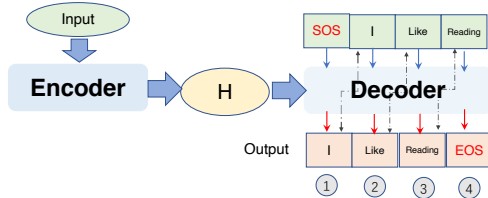

Figure 1: Working mechanism of NMT systems

remove the unimportant neurons. After pruning, the smaller size DNNs could achieve competitive accuracy with the original DNNs but require less computational costs. Another type of techniques Wang et al. (2018); Graves (2016); Figurnov et al. (2017), called input-adaptive techniques, this type of technique dynamically skip a certain part of the DNNs to reduce the number of computations in the inference time. By skipping some parts of the DNNs, the input-adaptive DNNs can balance the accuracy and computational costs. However, recent studies (Haque et al., 2020; Hong et al., 2020) show input-adaptive DNNs are not robustness against the adversary attack, which implies the input-adaptive will not save computational costs under attacks.

## 2.3 Adversarial Attacks

Existing work on adversarial machine learning has shown that even the state-of-the-art DNNs can be fooled by adversarial examples (Carlini & Wagner, 2017; Athalye et al., 2018). Adversarial examples are the elaborately crafted samples that apply human-unnoticeable perturbations on benign samples to maximize the target DNNs' errors. Based on prior knowledge about victim DNNs, the generation of adversarial examples could be categorized into white-box and black-box attacks. In the white-box settings (Goodfellow et al., 2014; Moosavi-Dezfooli et al., 2016; Kurakin et al., 2016; Carlini & Wagner, 2017; Jang et al., 2017; Madry et al., 2017; Chen et al., 2018; Rony et al., 2019), the adversary know the victim DNNs' architecture and parameters and can compute gradients to generate adversarial examples. In the black-box settings (Brendel et al., 2017; Chen et al., 2020; Brendel et al., 2019), the adversary can also exploit a surrogate model for launching the attack.

## 3 Attack Methodology

### 3.1 Threat Model

We consider an adversary who aims to decrease the efficacy of a victim NMT system. The attacker perturbs a benign sentence with unnoticeable perturbations to craft adversarial examples and feeds adversarial examples to the victim NMT. The perturbed adversarial examples will consume more computational resources of the victim NMT systems, thus impairing the translation service or make the service unavailable.

**Adversary's Capabilities.** The attacker is able to modify the victim's input samples to apply the perturbations, *e.g.,*, By publishing some documents contain adversarial examples on the Internet. We follow existing work (Cheng et al., 2020; Belinkov & Bisk, 2017; Li et al., 2018; Jin et al., 2020) and consider two types of perturbations: (i) token-level perturbation and (ii) character-level perturbation. To ensure imperceptibility, we limit the perturbation size $\epsilon$ that the adversary can perturb the benign inputs. In Section 4, we evaluate how different $\epsilon$ will affect the effectiveness of proposed attack.

**Adversary's Knowledge.** To assess the security vulnerability of existing NMT systems, we study white-box scenarios, *i.e.,* the attackers know the victim NMT systems architecture, parameters and the tokenizer that tokenize the word. In Section 4.4, we study more practical black-box scenarios,

*i.e.,*, the attacker leverage the transferability to generate adversarial examples and attack a victim NMT system without any prior knowledge about the victim.

**Adversary's Goals.** The adversary's goal is to decrease the efficacy of a victim NMT system. As we discussed in Section 2, NMT system's efficacy is related to its' output length. A longer output length indicates the NMT system costs more computational resources and becomes less efficacy. Thus, the adversary can achieve their goal through increasing the NMT system's output length.

$$\Delta = \text{argmax}_\delta \quad Len(\mathcal{F}(x + \delta)) \qquad s.t. \, ||\delta|| \leq \epsilon \qquad (1)$$

Finally, we formulate our problem of generating efficiency adversarial examples as an optimization problem. As shown in equation 1, where $x$ is the benign input, $\mathcal{F}$ is the target NMT system, $\epsilon$ is the maximum adversarial perturbation, and $Len(\cdot)$ measure the length of a output sequence. Our attack `TranSlowDown` tries to search a perturbation $\Delta$ that maximize the output length (decrease target NMT system $\mathcal{F}$ effiency) while smaller than the allowed perturbation (unnoticeable).

## 3.2 TRANSLOWDOWN ATTACK

Our attack algorithm is shown in Algorithm 1, which iteratively performs the following three main steps: (*i*) find important tokens, (*ii*) generate possible perturbation, and (*iii*) select optimal perturbation, until the generated adversarial examples reach the maximum perturbation.

**Find Important Tokens (line 3 to 6):**

Given a benign input $x = [tk_1, \cdots, tk_m]$, the first step is to find each tokens importance to NMT systems' efficiency. As we discussed in Section 2, NMT systems' efficiency is related to its' output length, and the output length is determined by the probability of EOS tokens. Then our objective is to decrease the probability of the EOS token to reduce NMT's efficiency. Formally, let NMT system's output probability be a sequence of vectors, *i.e.*, $\mathcal{F}(x) = [p_1, p_2, \cdots, p_n]$. Then the probability of EOS tokens are $[p_1^{eos}, p_2^{eos}, \cdots, p_n^{eos}]$. We seek to find the importance of each token $tk_i$ in $x$ to this probability sequence. We also observe that the output token sequence will affect EOS's probability. Thus, we define the importance score of token $tk_i$ as $g_i$, shown in equation 2.

---

**Algorithm 1** `TranSlowDown` Attack

**Input:** Benign input $x$, victim NMT system $\mathcal{F}(\cdot)$, maximum perturbation $\epsilon$
**Output:** Adversarial examples $x'$

1: $x' \Leftarrow x$  Initialize $x'$ with $x$
2: **while** $||x' - x|| \leq \epsilon$ **do**
3:     **for** each $tk_i \in x'$ **do**
4:         Compute $g_i$ according to equation 2
5:     **end for**
6:     TK $\Leftarrow Sort(tk_1, tk_2, \cdots, tk_m)$ according to $g_i$
7:     $L = \text{GeneratePerturbation}(TK, x', \mathcal{F}(\cdot))$
8:     $[tk, \Delta] = \text{SelectBestPerturbation}(L, x', \mathcal{F}(\cdot))$
9:     $x' \Leftarrow$ replace $tk$ with $\Delta$ in $x'$
10: **end while**
11: return $x'$

---

$$o_i = \text{argmax}(p_i) \qquad f(x) = \frac{1}{n}\sum_i^n (p_i^{eos} + p_i^{o_i}) \qquad g_i = \sum_j \frac{\partial f(x)}{\partial tk_i^j} \qquad (2)$$

Where $[o_1, o_2, \cdots, o_n]$ is the current output token, $f(x)$ is the probability we seek to minimize, $tk_i^j$ is the $j^{th}$ dimension of $tk$'s embeddings, and $g_i$ is the derivative of $f(x)$ to $i^{th}$ token's embedding.

**Adversarial Perturbation Generation (line 7):** After identifying important tokens, the next step is to mutate the important token with unnoticeable perturbations. In this step, we get a set of perturbation candidate $L$ after we perturbing the most important tokens in the original input. Following existing work, we consider two kinds of perturbations, *i.e.*,, token-level perturbation and character-level perturbation. Table 1 shows some example of token-level and character-level perturbation under different perturbation size $\epsilon$, we color the perturbation with color red.

$$\mathcal{I}_{src,tgt} = \sum_j (E(tgt) - E(src)) \times \frac{\partial f(x)}{\partial tk_i^j} \qquad \delta = \text{argmax}_{tgt}\, \mathcal{I}_{tk,tgt}; \qquad (3)$$

For token-level perturbation, we consider replacing the original token $tk$ with another token $\delta$. To compute the target token $\delta$, we define token replace increment $\mathcal{I}_{src,tgt}$ to measure the efficiency degradation of replacing token $src$ to $tgt$. As shown in equation 3, $E(\cdot)$ is the function to get corresponding token's embedding, $E(tgt) - E(src)$ is the vector increment in embedding space. Because $\frac{\partial f(x)}{\partial tk_i^j}$ indicate the sensitive of output length to each embedding dimension, $\mathcal{I}_{src,tgt}$ represent the total benefits of replacing token $src$ with $tgt$. We search the target token $\delta$ in the vocabulary to maximize the token replace increment with the source token $tk$.

For character-level perturbation, we consider character insert perturbation. Specifically, we insert one character $c$ into token $tk$ to get another token $\delta$. The character-inset perturbation is common in the real world when typing quickly and is unnoticeable without careful examination. Because character insertion is likely to result in out-of-vocabulary (OOV), thus it is challenging to compute the token replace increment as token-level. Instead, we enumerate possible $\delta$ after character insertion to get candidate set $L$. Specifically, we consider all letters and digits as the possible character $c$ because humans can type these tokens through the keyboard, and we consider all positions the possible insertion position. Then for token $tk$, which contains $l$ characters, there are $(l + 1) \times ||C||$ perturbation candidates, where $||C||$ is the size of all possible characters.

Table 1: Examples of token-level and character-level perturbation under different size

| Original | | $\epsilon$ | Do you know who Rie Miyazawa is? |
|---|---|---|---|
| Token-Level | | 1 | Do I know who Rie Miyazawa is? |
| | | 2 | Do I know who Hill Miyazawa is? |
| | | 3 | How I know who Hill Miyazawa is? |
| Character-Level | | 1 | Do you know who Rie Miya-zawa is? |
| | | 2 | Do you know whoo Rie Miya-zawa is? |
| | | 3 | Do you knoiw whoo Rie Miya-zawa is? |

**Select Best Perturbation (line 8 to 9):** After collecting candidate perturbations $L$, we select an optimal perturbation from the collected candidate sets. As our objective is searching adversarial perturbation that will produce longer output length, thus, we try all adversarial perturbation and select the optimal perturbation that produce the maximum output length.

## 4 EVALUATION

### 4.1 EXPERIMENTAL SETUP

**Models and Datasets.** As shown in Table 2, we consider the following three public NMT systems as our victim models: Google's T5 (Raffel et al., 2019), Facebook's Fairseq Transformer (Ng et al., 2019), and Helsinki-NLP's H-NLP Translator (Jörg et al., 2020). For each model, we choose a specific testing dataset: *(i).* T5 is released by Google, it's first pre-trained with multiple language problems, then fine-tuned on the English-German translation task. We apply English sentences from dataset ZH19 as benign examples to generate adversarial examples; *(ii).* Fairseq is one of the NMT models that Facebook FAIR submitted to the WMT19 shared news translation task, and it's based on the FFN transformer architecture (Vaswani et al., 2017). We select Fairseq's en-de model as our victim model, which is designed for the English-German translation task. We apply WMT19 test data as benign examples to generate adversarial examples; *(iii).* H-NLP is a seq2seq translation model, the source language is English and the target language is Chinese. We apply the sentences from Tatoeba-test.eng.zho as benign examples to generate adversarial samples.

Table 2: The victim models in our experiments

| Model | Vocab Size | Source | Target | URL |
|---|---|---|---|---|
| T5 | ZH19 | En | De | https://huggingface.co/t5-small |
| FAIR | WMT19 | En | De | https://huggingface.co/facebook/wmt19-en-de |
| H-NLP | Tatoeba | En | Zh | https://huggingface.co/Helsinki-NLP/opus-mt-en-de |

**Metrics.** We apply floating-point operations (FLOPs) and response latency to measure victim NMT systems' efficiency. FLOPs is a hardware-independent metric and is widely used to measure DNNs' computational complexity. Higher FLOPs mean that the DNN requires more computations to handle an input, which represents less efficiency (Howard et al., 2017; Zhang et al., 2018). Response latency is a hardware-dependent metric, which can measure the victim model's efficiency on benign and adversarial examples. High response latency indicates that the victim NMT system needs to spend

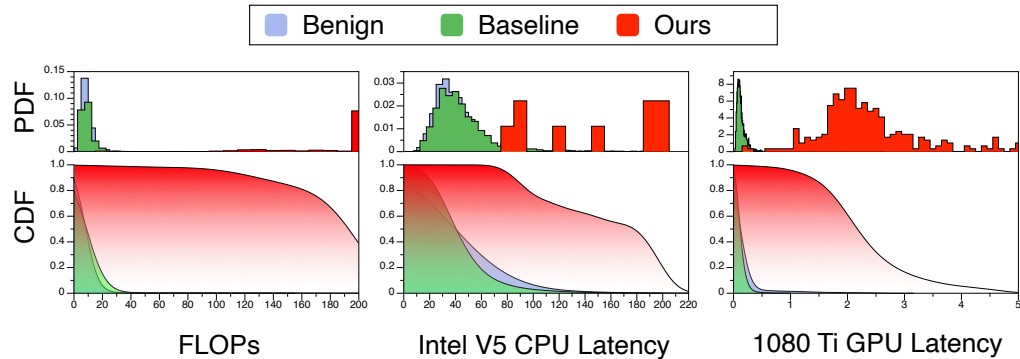

Figure 2: The distribution of FLOPs and latency before and after token-level attacks

more computational resources. The higher the response latency, the worse the real-time translation quality. We measure the latency on two hardware platforms: Intel Xeon E5-2660v3 CPU and Nvidia 1080Ti GPU.

To evaluate the effectiveness of `TranSlowDown`, we measure the FLOPs and latency on benign and adversarial examples respectively. Specifically, we first compute the probability density function (PDF) and cumulative distribution function (CDF) of FLOPs and latency. Then, we define two metrics: I-FLOPs and I-Latency, to evaluate how the adversarial samples affect the victim NMT systems. The formal definition of I-FLOPs and I-Latency shows in equation 4, where $x$ denotes the benign example and $x'$ represents the adversarial example after perturbing $x$, FLOPs($\cdot$) and Latency($\cdot$) are the functions to calculate the average FLOPs and latency per input character. Higher I-FLOPs and I-Latency indicate a more severe slowdown caused by the adversarial example.

$$\text{I-FLOPs} = \frac{\text{FLOPs}(x') - \text{FLOPs}(x)}{\text{FLOPs}(x)} \times 100\% \quad \text{I-Latency} = \frac{\text{Latency}(x') - \text{Latency}(x)}{\text{Latency}(x)} \times 100\%$$

(4)

**Comparison Baseline.** To the best of our knowledge, we are the first to study the attack efficiency of NMT systems, therefore no existing efficiency attack framework can be applied as the baseline. To show existing accuracy-based attacks can not be applied for evaluating NMT's efficient robustness, we compare `TranSlowDown` against two accuracy-based attacks. We choose Seq2Sick (Cheng et al., 2020) as the token-level baseline. Seq2Sick can replace the tokens in benign inputs to produce adversarial translation outputs that are entirely different from benign outputs. Because Seq2Sick only works under token-level, we choose SyntheticError (Belinkov & Bisk, 2017) as the character-level baseline. SyntheticError minimizes the NMT system's accuracy (BLUE score) by introducing synthetic noise. Specifically, SyntheticError introduces four character-level perturbation: Swap, Middle Random, Fully Random, and Keyboard Typo to perturb benign examples to decrease the NMT system's BLUE scores.

### 4.2 TOKEN-LEVEL PERTURBATION RESULTS

**Effectiveness of Token-level Attack.** Figure 2 shows the distribution of H-NLP's efficiency metrics under token-level (more results for T5 and Fairseq can be found in Appendix A.2). The first and second rows show the PDF and CDF [1] results respectively. In each plot, the blue area denotes the distribution of benign examples, green and red areas represent the distribution of adversarial examples generated from the comparison baseline Seq2Sick and `TranSlowDown` respectively. It can be clearly observed from Figure 2 that adversarial examples generated by `TranSlowDown` significantly change the distribution of the victim NMT system's FLOPs and latency. In contrast, the adversarial examples from baseline have very little effect on NMT's efficiency. The results for the other two NMT systems in the Appendix show similar trends with Figure 2. From the results, we conclude that: existing accuracy-based attacks can not be applied to evaluate the efficiency robustness of NMT systems. In contrast, our proposed attack, `TranSlowDown`, effectively generates

---

[1]For better presentation, we plot the CDF from one to zero.

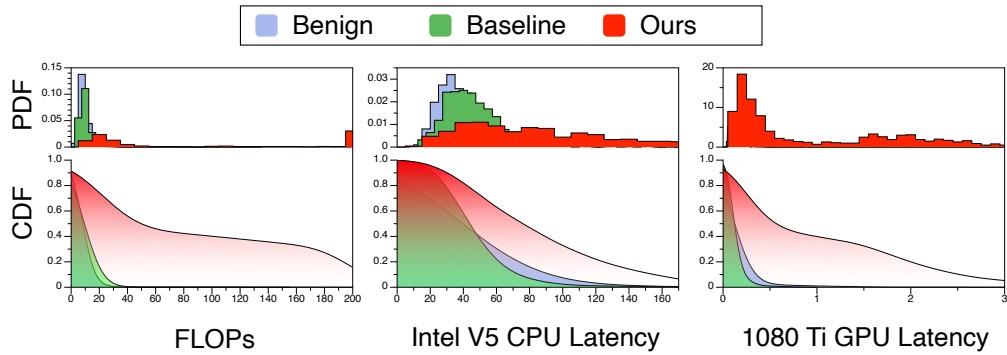

Figure 3: The distribution of FLOPs and latency before and after character-level attacks

adversarial examples to slow down NMT systems. Thus, `TranSlowDown` is effective in evaluating the efficiency robustness of NMT systems.

Table 3: The severity of token-level adversarial attacks

| NMT | Perturbation | I-FLOPs | | I-Latency (CPU) | | I-Latency (GPU) | |
|---|---|---|---|---|---|---|---|
| | | Baseline | Ours | Baseline | Ours | Baseline | Ours |
| **H-NLP** | 1 | 4.18 | **1318.72** | 4.51 | **1485.64** | 2.55 | **1269.53** |
| | 2 | 6.03 | **2131.92** | 6.33 | **2403.21** | 3.05 | **2054.83** |
| | 3 | 12.94 | **2336.88** | 13.83 | **2636.51** | 11.10 | **2296.09** |
| | 4 | 17.57 | **2360.94** | 19.40 | **2664.46** | 15.68 | **2314.23** |
| | 5 | 23.56 | **2366.09** | 25.38 | **2669.82** | 23.89 | **2321.93** |
| **FairSeq** | 1 | -0.05 | **23.46** | 0.15 | **24.85** | 3.50 | **29.62** |
| | 2 | -0.19 | **36.80** | 0.61 | **38.97** | 1.17 | **40.59** |
| | 3 | -2.22 | **47.27** | -0.73 | **51.21** | -1.40 | **54.44** |
| | 4 | -5.78 | **57.60** | -4.17 | **63.35** | -4.09 | **63.57** |
| | 5 | -8.20 | **80.75** | -6.85 | **89.73** | -1.76 | **85.74** |
| **T5** | 1 | 10.09 | **335.41** | 9.66 | **383.38** | 7.87 | **352.88** |
| | 2 | 6.80 | **343.69** | 5.44 | **393.20** | 4.06 | **362.45** |
| | 3 | 1.47 | **343.69** | 1.64 | **393.20** | -1.54 | **362.45** |
| | 4 | -11.66 | **343.69** | -9.62 | **393.20** | -8.28 | **362.45** |
| | 5 | -25.18 | **343.69** | -24.29 | **393.20** | -8.59 | **362.45** |

**Severity of Token-level Attack.** To quantify the severity of the proposed efficiency attack, we measure I-FLOPs and I-Latency under different perturbation sizes. From the results in Table 3, we have the following observations: *(i)* For all experimental subjects, the adversarial examples generated by `TranSlowDown` slow down the victim NMT systems by a large margin compared to the baseline method. For H-NLP, `TranSlowDown` adversarial examples increase the FLOPs, CPU latency, GPU latency up to 2366.09%, 2669.82%, 2321.93% respectively. *(ii)* As perturbation size increases, the adversarial examples generated by `TranSlowDown` can make the victim NMT system slower. However, this observation does not hold for the baseline method. This is because the baseline method is designed to decrease victim NMT's accuracy, increasing adversarial perturbation may decrease NMT accuracy but does not imply to consume more computational resources. *(iii)* The adversarial examples generated by the baseline method can not always ensure increasing FLOPs and latency, while `TranSlowDown` adversarial examples increase FLOPs and latency on all settings. This result is consistent with the second observation and indicates that the baseline method is not suitable to evaluate NMT's computational consumption robustness.

## 4.3 CHARACTER PERTURBATION

**Effectiveness of Character-level Attack.** Figure 3 shows the distribution of H-NLP's efficiency metrics under character-level (more results for T5 and Fairseq can be found in Appendix A.2). For character-level attack, compared with baseline, adversarial examples generated by `TranSlowDown`

significantly change the distribution of the victim NMT system's FLOPs and latency as well. The results for the other two NMT systems in the Appendix show similar trends with Figure 3. From the results, we obtain a similar conclusion with the token-level attack, *i.e.,*, our proposed attack, `TranSlowDown`, effectively generates adversarial examples to slow down NMT systems. In contrast, the accuracy-based attack can not achieve this goal.

Table 4: The severity of character-level adversarial attacks

| NMT | Perturbation | I-FLOPs | | I-Latency (CPU) | | I-Latency (GPU) | |
|---|---|---|---|---|---|---|---|
| | | Baseline | Ours | Baseline | Ours | Baseline | Ours |
| H-NLP | 1 | 20.09 | **389.36** | 21.27 | **431.72** | 11.84 | **368.08** |
| | 2 | 20.09 | **879.16** | 21.29 | **978.47** | 12.07 | **840.47** |
| | 3 | 20.09 | **1102.86** | 21.29 | **1232.71** | 12.07 | **1056.88** |
| | 4 | 20.09 | **1189.48** | 21.29 | **1328.29** | 12.07 | **1136.70** |
| | 5 | 20.09 | **1224.91** | 21.29 | **1366.22** | 12.07 | **1174.57** |
| FairSeq | 1 | 0.28 | **36.23** | 0.71 | **38.59** | 3.15 | **40.13** |
| | 2 | 0.28 | **87.92** | 0.71 | **97.37** | 3.17 | **94.31** |
| | 3 | 0.28 | **145.60** | 0.71 | **164.90** | 3.17 | **155.43** |
| | 4 | 0.28 | **190.43** | 0.71 | **217.94** | 3.17 | **204.02** |
| | 5 | 0.28 | **223.07** | 0.71 | **255.42** | 3.17 | **235.93** |
| T5 | 1 | 5.60 | **217.46** | 6.39 | **249.44** | 5.71 | **229.70** |
| | 2 | 5.56 | **249.88** | 6.37 | **286.81** | 5.69 | **258.35** |
| | 3 | 5.56 | **267.58** | 6.37 | **307.57** | 5.69 | **273.41** |
| | 4 | 5.56 | **276.33** | 6.37 | **318.40** | 5.69 | **283.37** |
| | 5 | 5.56 | **280.10** | 6.37 | **323.67** | 5.69 | **286.84** |

**Severity of Character-level Attack.** Similar to Section 4.2, Table 4 presents the severity of character-level adversarial examples. The results show consistency with the token-level attack. For the character-level attack, the adversarial examples generated by `TranSlowDown` also significantly slow down the victim NMT systems compared to the baseline method. For H-NLP, `TranSlowDown` adversarial examples increase the FLOPs, CPU latency, GPU latency up to 1224.91%, 2669.82%, 1174.57% respectively. As perturbation size increases, the adversarial examples generated by `TranSlowDown` can make the victim NMT system slower, which is not reflected in the baseline results. This is consistent with the token-level attack.

## 4.4    THE TRANSFERABILITY OF ADVERSARIAL EXAMPLES

In this section, we study the transferability of efficacy adversarial examples. Even though white-box attacks are important to expose the vulnerability, black-box attacks are more practical in the real world because they require less information about the NMT systems. We investigate whether `TranSlowDown` is transferable under black-box settings.

Table 5: The maximum I-FLOPs of blackbox token-level attack

| Source | Target | 1 | 2 | 3 | 4 | 5 |
|---|---|---|---|---|---|---|
| H-NLP | FairSeq | 185.71 | 57.14 | 68.75 | 57.14 | 42.86 |
| | T5 | 1600.00 | 1600.00 | 1600.00 | 1600.00 | 1600.00 |
| FairSeq | H-NLP | 6566.67 | 6566.67 | 6566.67 | 6566.67 | 6566.67 |
| | T5 | 1600.00 | 1600.00 | 1600.00 | 1600.00 | 1600.00 |
| T5 | H-NLP | 3733.33 | 3733.33 | 3733.33 | 3733.33 | 3733.33 |
| | FairSeq | 858.33 | 1816.67 | 1816.67 | 945.45 | 945.45 |

Specifically, we treat one NMT system as a target and apply another NMT system as the source to generate adversarial examples. We then feed the adversarial examples to the target NMT systems and measure the maximum I-FLOPs. Maximum I-FLOPs indicate the efficiency degradation under the worst scenario, which is important to measure the vulnerability of NMT systems. Notice the source and the target NMT systems in our experiment adopts different model architectures and are trained with different datasets. Thus, if the adversarial examples can increase the FLOPs of the target NMT system, it proves that transferability exists in our attack. The results for token-level attacks are shown in Table 5 (more results in Appendix A.3). From the results, we observe that for

all experimental settings, `TranSlowDown` generates adversarial examples that increase the target NMT system's computational FLOPs to a large extend. The results imply that under the worst scenarios, the attackers can generate efficient adversarial examples even without prior knowledge about the victim NMT systems.

## 5 REAL WORLD CASE STUDY

We conduct a case study to evaluate `TranSlowDown`'s ability to attack real world mobile devices' battery power.

**Experiment Settings.** We select Google's T5 as our victim NMT model. We first deploy the model on Samsung Galaxy S9+, which has 6GB RAM and a battery capacity of 3500 mAh. After that, we select one sentence from the dataset `ZH19` as our testing example; We then apply `TranSlowDown` to perturb the benign

Table 6: Sentences for energy attack on mobile devices

| | |
|---|---|
| **Benign** | Death comes often to the soldiers and marines who are fighting in anbar province, which is roughly the size of louisiana and is the most intractable region in iraq. |
| **Adversarial** | Death comes often to the soldiers and marines who are fighting in anbar province, which is roughly the **(size** of of louisiana and is the most intractable region in iraq. |

example with character-level perturbation and obtain the corresponding adversarial example. The benign sentence and the adversarial sentence are shown in Table 6 (More results in Appendix section A.4), we color the perturbation with the color red. Notice the adversarial example insert only one character in the benign sentence. This one-character perturbation is very common in the real world because of the user's typos. Finally, we feed the benign and adversarial examples to the deployed NMT system and measure the NMT's energy consumption of translating benign and adversarial examples.

**Experiment Results.** The mobile's battery power change is shown in Figure 4. The red line is for adversarial example, and the blue line is for benign example. The results show that the adversarial example increases the mobile device's battery power significantly more significant than the benign example. Specifically, after 300 iterations, the adversarial example consumes 30% of the battery power, while the benign example only consumes less than 1%. The results show the vulnerability of the efficiency attack for IoT and mobile devices. Recall that the adversarial example used in our experiment only inserts one character in the benign sentence. This minimal perturbation can result from benign user typos instead

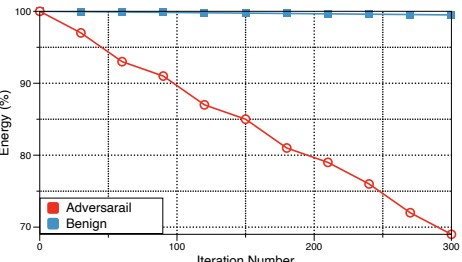

Figure 4: Remaining battery power of the mobile device after T5 translating benign and adversarial sentence

of from the adversary. Thus, the results suggest the criticality and the necessity of increasing NMT systems' efficiency robustness.

## 6 DISCUSSION

Although there are many accuracy-based defense mechanisms to protect DNNs. However, accuracy-oriented defense mechanisms are not applicable for protecting NMT systems' efficiency robustness because of two reasons: (i) Many existing accuracy defense mechanisms require running the model with the input and use the hidden states of the model to make decision. However, running the input would defeat the purpose of an energy defense as the computation resource is already consumed. (ii) Our results in section 4 show that accuracy-based adversarial examples and efficiency-based adversarial examples belong to different distribution, thus, running accuracy-based detector can not detect efficiency-based adversarial examples successfully.

## 7 CONCLUSIONS

In this work, we study the robustness of NMT systems against adversarial efficiency attacks. Specifically, we propose `TranSlowDown`, an attack that introduces imperceptible adversarial perturbations to benign inputs to increase NMT's computational complexity. We evaluate `TranSlowDown` on three public available NMT systems, the results show `TranSlowDown` generates adversarial examples decrease NMT systems' efficiency. Our study suggests that efficiency attacks are a vulnerable, yet under-appreciated, threat against NMT systems.

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

# A APPENDIX

## A.1 TOKEN-LEVEL RESULTS

Figure 5 and Figure 6 show the distribution of T5 and Fairseq's efficiency metrics under token-level respectively. Similar to Section 4.2, adversarial examples generated by `TranSlowDown` significantly change the distribution of the victim NMT system's FLOPs and latency while the adversarial examples from baseline have minor effect on NMT's efficiency.

## A.2 CHARACTER-LEVEL RESULTS

Figure 7 and Figure 8 show the distribution of T5 and Fairseq's efficiency metrics under character-level respectively. Similar to Section 4.3, compared with the baseline, the adversarial samples generated by `TranSlowDown` still achieved higher FLOPs and latency.

## A.3 TRANSFERABILITY RESULTS

As we mentioned in Section 4.4, we investigate whether `TranSlowDown` is transferable under black-box settings. The results for character-level attacks are shown in Table 7, which is consistent with the result obtained in the token-level experiment. The results further prove that the attacker can generate effective adversarial examples even without prior knowledge about the victim NMT system.

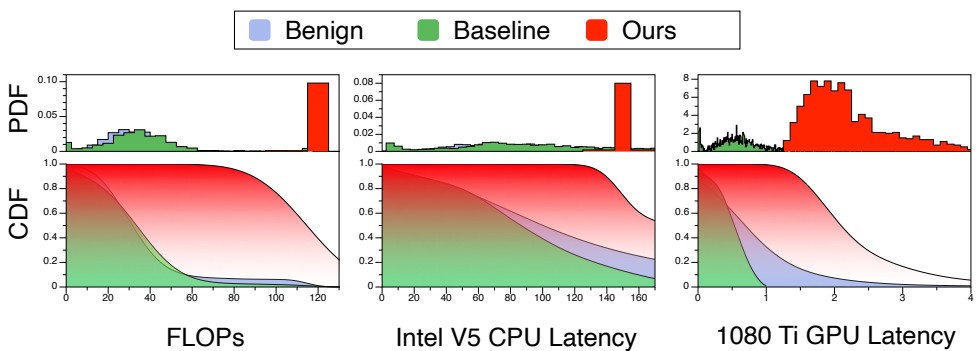

Figure 5: The efficiency metric of T5 before and after token-level attacks

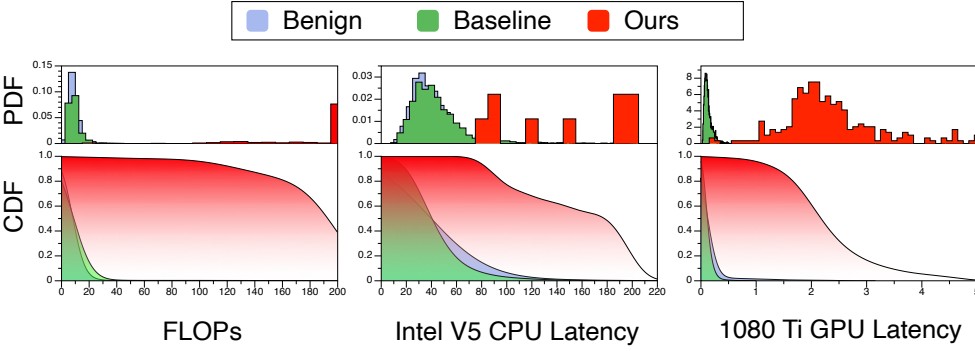

Figure 6: The efficiency metric of FairSeq before and after token-level attacks

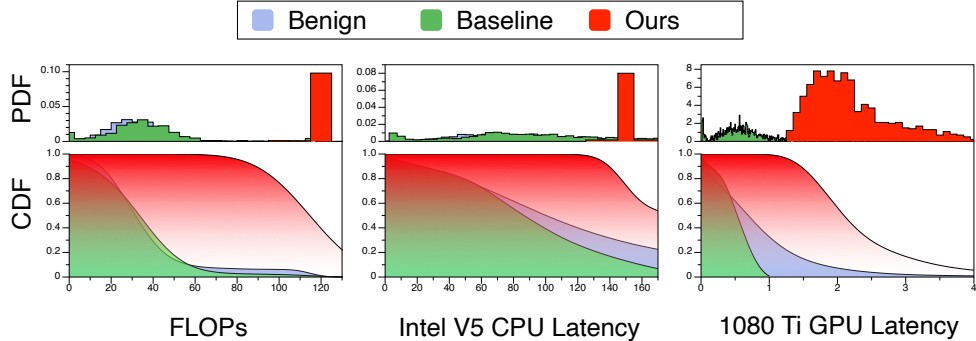

Figure 7: The efficiency metric of T5 before and after character-level attacks

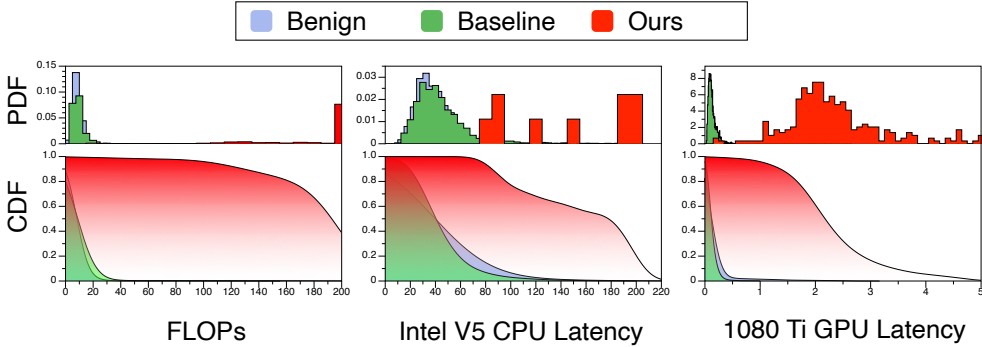

Figure 8: The efficiency metric of FairSeq before and after character-level attacks

Table 7: The maximum I-FLOPs of blackbox character-level attack

| Source | Target | 1 | 2 | 3 | 4 | 5 |
|--------|--------|-----|-----|-----|-----|-----|
| **H-NLP** | **FairSeq** | 900.00 | 273.08 | 2400.00 | 573.08 | 2400.00 |
| | **T5** | 1700.00 | 1700.00 | 1700.00 | 1700.00 | 1700.00 |
| **FairSeq** | **H-NLP** | 250.00 | 325.00 | 400.00 | 400.00 | 433.33 |
| | **T5** | 1400.00 | 1400.00 | 1400.00 | 1400.00 | 1400.00 |
| **T5** | **H-NLP** | 3733.33 | 3733.33 | 3733.33 | 3733.33 | 3733.33 |
| | **FairSeq** | 1177.78 | 1337.50 | 1542.86 | 1337.50 | 1337.50 |

## A.4 CASE STUDY RESULTS

In this section, we put more generated efficiency adversarial examples for all our victim NMT systems.

Table 8: The benign and adversarial examples under different settings

| Attack Type | NMT | Benign | Adversarial 1 |
|---|---|---|---|
| Token | H-NLP | Let me see. | 哎 me see |
| | | I am here. | Iam going. |
| | | He is back. | He is 视 |
| | FairSeq | configure, verify, and troubleshoot vtp | configure, bebelus, and troubleshoot vtp |
| | | finger-pointing politicians and chest-beating nationalists | classmates-pointing politicians and chest-beating nationalists |
| | | in the two nations will make rational discussion nearly impossible | in the two nations will make rational discussion nearly impossible. |
| | T5 | The Commission's report sets out a detailed analysis of all the replies. | The Commission's report sets out a detailed analysis of all the replies Gefahr |
| | | The October European Council meeting will return to the issue of migration. | The October European Council meeting will return to the issue of migration not |
| | | on z/os the maximum length of the attribute is 256 bytes. on all other platforms it is 1024 bytes. | on z presenceos the maximum length of the attribute is 256 bytes. on all other platforms it is 1024 bytes. |
| Character | H-NLP | You have heart disease. | You have heart Odisease. |
| | | This is my question. | ThWis is my question. |
| | | I'm a little confused. | I'm a litt5le confused. |
| | FairSeq | President Juncker: 'Europe needs a genuine security union' | President Juncker: 'Europe needs a genuine security uni(on '" |
| | | step up the cooperation on return and readmission. | step up the cooperation on return and readmis0sion. |
| | | In the refugee crisis, Turkey and the EU walk together and work together. | In Ythe refugee crisis, Turkey and Ythe EU walk togeYther and work togeYther. |
| | T5 | load, performance, or scalability | load, perf'ormance, "or scalability |
| | | will hell swallow you up? | will hell swallow $you up? |
| | | shengli oilfield used euronavy es301 to make coatings to offshore tidal zone. | shengli oilfield used euronavy es301 to make coatings to offshore tidal zone. |

