# OpenReview forum: "TransSlowDown: Efficiency Attacks on Neural Machine Translation Systems"
_ICLR.cc/2022/Conference — ICLR 2022 Submitted_

### Official Review · Reviewer_cSUu · 2021-10-26

**Correctness:** 3
**Technical Novelty And Significance:** 4
**Empirical Novelty And Significance:** 3
**Recommendation:** 6
**Confidence:** 4

**Details Of Ethics Concerns:**

The paper discusses an attack on NMT efficiency. I *don't* think that it should be flagged as "potentially harmful insight" as using the described method to attack real world systems seems sort of far fetched to me, but having an additional pair of eyes cannot hurt.

**Main Review:**

The robustness of NMT against adversarial input sentences has not received much attention (if any) in the literature. Raising this question and demonstrating an effective attack would be a valuable contribution to the community. The demonstrated impact of TransSlowDown seems indeed impressive first, the paper is clearly written, but I feel that at times it is (perhaps unconsciously) on the verge of overselling the results.

First, all NMT systems in practice use a maximum sentence length, which is usually set to a factor times the source sentence length. This maximum target sentence length is needed to prevent infinite loops when NMT is hallucinating and to avoid OOMs. This factor is usually 2-3. Fig. 2 suggests that this factor was either not used or was set to a very high value. Since TransSlowDown works by increasing the target sentence length, specifying and/or varying this factor is crucial for assessing the results.

Second, T5 and fairseq are pure Transformer models with Transformer decoders. Using self-attention in the decoder results in an inference time complexity of n^2. Therefore, an attack like TransSlowDown that increases the target sentence length is expected to be particularly effective. Most deployed MT systems use recurrency instead of self-attention in the decoder for efficiency reasons, so it would be worth to test the effectiveness of TransSlowDown on architectures with LSTM/GRU decoders.

I still think that the paper has merits, it just needs to be more upfront about the details and the limitations of the results.

Some pointers to related work are missing. The literature on faithfulness in NMT seems relevant, and in Sec. 2.2 at least two very common techniques to improve the efficiency of neural networks (quantization and distillation) are not cited.

Minor comments
- i in Sec. 3.2 sometimes runs over the input sentence [1,m], sometimes over the output sentence [1,n]
- Minor typos throughout the paper

**Summary Of The Paper:**

TransSlowDown is an attack scheme that aims to targets the computational resources of NMT systems. The authors show that small modifications to benign input sentences can significantly increase the computation NMT needs to process the input, paving the way for denial-of-service attacks on translation services or attacks on the battery power of mobile devices. TransSlowDown is a white box technique that makes use of gradients of the target NMT system, but can also be adapted to a more practical block box technique where gradients from another NMT system are used.

**Summary Of The Review:**

The paper explores an important aspect - efficiency robustness of NMT systems - that is clearly understudied in the literature. Some missing details and experimental setups prevent me from giving a stronger acceptance recommendation.

---

> ### Author Response · Authors · 2021-11-23
> **Uploaded Rebuttal**
>
> We thank the reviewer for the feedback on our work. The response to all the queries has been uploaded as an additional document. The General Response is for the queries asked by multiple reviewers. Following that, we address the queries of each specific reviewer.
>
> 1. Configuration of the Maximum Length: we apply the default maximum length of the victim models, which we downloaded form  HuggingFace. We add another experiment, where the maximum length is set according to the input length. The results are shown in the link of the general response. The results show our attacks can also slow down the NMT models under these settings.
>
> 2. Evaluation on Other Model Architectures: We evaluate our proposed attacks on two more model architectures (LSTM and GRU encoder and decoder). The results are listed in the general response link.

---

### Official Review · Reviewer_Rkum · 2021-11-01

**Correctness:** 3
**Technical Novelty And Significance:** 2
**Empirical Novelty And Significance:** 2
**Recommendation:** 5
**Confidence:** 4

**Main Review:**

Strengths:
- The idea of attacking the computation complexity of NMT model is interesting and practical in industrial applications.
- The paper is well-written and easy to follow.

Weaknesses:
- The paper considers the white-box attacking scenario, which is less significant than black-box attacking which fits the real-world application more. In my view, the significance of white-box attacking is to find the vulnerable point of the model and then fix it by some corresponding solutions such as adversarial training/regularization loss function. The paper only proposes the attacking method without equipping it with the fixing method, which limits its contribution. Therefore I suggest the authors add this part to show that the proposed method can increase the robustness of the NMT model w.r.t the computation complexity attacking.
- In textual adversarial attacking, a very important condition is that the generated adversarial sample should be imperceptible with the input, in human knowledge. Therefore an indispensable part in related papers is to prove this point, which is missed in this paper, however. I suggest the authors conduct evaluations between the similarity of the source input as well as the generated adversarial sample, in automatic (such as BLEU) and human evaluation (such as grammar/semantic/consistency) metrics.
- The proposed method seems time consuming as nearly all operations are based on greedy search. How long does the method take in experiments? Have you compare the running time with other baselines?

Minor points:
- The insight behind of Equ (2) is not introduced. How the importance is defined? What does each part represents? Needs more explantion.
- In character-level perturbation, if the corrupted result is tokenized into multiple tokens or the UNK token, how to maintain them are still the most important tokens? As the probability may change if the source length is changed.
- How to maintain the corrupted result is in the mamximum permutation \epsilon?

**Summary Of The Paper:**

The paper proposes an adversarial attack method for NMT, aiming to increase the computation complexity while decoding by introducing perturbations into the input. To achieve that, the authors propose an attacking algorithm that consists of three steps: find the most important source token w.r.t the probability of EOS token, then add token level (by FGSM) or character level perturbations (by greedy search), and finally select the perturbation that produces the longest result.

**Summary Of The Review:**

The paper proposes an interesting idea of a new attacking target of NMT models, but has some major flaws such as limited contribution and inadequate evaluation. Therefore I vote for a rejection.

---

> ### Author Response · Authors · 2021-11-23
> **Uploaded Rebuttal**
>
> We thank the reviewer for the feedback on our work. The response to all the queries has been uploaded as an additional document. The General Response is for the queries asked by multiple reviewers. Following that, we address the queries of each specific reviewer.
>
> 1. Similarity of the Source Inputs and the Translated Outputs:
> We measure the similarity of benign inputs and adversarial inputs using the BLEU-4 score. The BLEU-4 scores are listed in the link of general response.
>
> 2. Overhead of the Attack Algorithms:  Our attack algorithm does not cost many overheads, because it only iterates a limited number of times~(the iteration number is equal to the maximum perturbation size).  Although we mutate the benign examples to generate some adversarial candidates in each iteration,  we batch the generated candidates to select an optimal one with the help of GPU parallelism. The overhead of this process is only a little larger than the overhead of querying the victim NMT models.
> The overhead results are listed in the link of general response.
>
> 3. Insight of Equation 2: The intuition of Eq 2 comes from two perspectives: \textit{(i)} The goal of the efficiency adversarial examples is to increase the output length to waste the computational consumption. The output length of NMT models is determined by the likelihood of EOS tokens, thus, our first objective is to decrease the likelihood of EOS in order to delay the appearance of EOS. The first objective can be formulated as minimize $ \frac{1}{n}\sum_{i}^{n} p_i^{eos}$
> \textit{(ii)} At the beginning of the optimization, $[p_1^{eos}, p_2^{eos}, \cdots, p_{n-1}^{eos}]$ are usually small while $p_{n}^{eos}$ is large. So $[o1, o_2, \cdots, o_{n-1}]$ keep the same at the beginning of the optimization.
> However, the process of NMT models generate output tokens is a Markov process $p_i = F_{decoder}(o_{i-1}, h)$.
> If $o_{n-1}$ keeps the same, modify the inputs only affect $h$, to accelerate the optimization process, we seek to modify leave the original output $[o_1, o_2, \cdots, o_{n-1}]$. The second objective can be formulated as minimize $ \frac{1}{n}\sum_{i}^{n} p_i^{o_i}$. Combining the above two objectives, we have the final objective function in Eq 2.
>
> 4. Ensure the Adversarial Perturbation less than the Maximum Perturbation: At each iteration, we mutate the input with one perturbation size. Thus, the iteration number will limit the adversarial perturbation size. If the maximum perturbation size is 5, then we just iterative 5 loops.
>
> 5. UNK Related Issues: It is okay for our algorithm if the corrupted result is tokenized into multiple tokens or the UNK token. Because for the next iteration, our algorithm will apply Equation 2 to find the new important tokens in the mutated sentences. If the UNK token is the most important token in the new sentences, our algorithm will mutate the UNK token and select an optimal candidate.

---

### Official Review · Reviewer_zCy9 · 2021-11-02

**Correctness:** 2
**Technical Novelty And Significance:** 3
**Empirical Novelty And Significance:** 3
**Recommendation:** 5
**Confidence:** 5

**Main Review:**

***Strengths***
1. The paper is easy to follow and the motivation is very clear.
2. The topic is interesting and can attract much attention from industry research.
3. The proposed method is intuitive and well-designed. The part of finding important tokens might benefit other research lines such as the over- and under-translation of NMT.

***Weaknesses***
1. In the paper, the authors repeatedly mention the relationship between NMT efficiency and output length. So, why not show the change of output length after the adversarial attacks? I think the sentence length is a more direct metric to evaluate NMT efficiency and is hardware-independent.
2. A very simple solution to the attack is applying some constraints to the beam search process of NMT. For example, terminating the decoding when the output is twice the length of the input (please refer to --max-len-b of fairseq-generate). If a simple constraint can solve the issue, the contribution of the paper will be significantly decreased.
3. Since the proposed method is more relevant to practical NMT systems, so, how about the results (Section 4.4) on Google translator or other online systems?

**It would be nice if the authors could add some experimental results of the above weaknesses during the author response.**

**Summary Of The Paper:**

This paper focuses on the efficiency of neural machine translation (NMT) systems, proposing a novel attack approach to test the efficiency robustness. The attack approach can be divided into three parts: 1) find the most relevant tokens of inference efficiency; 2) generate adversarial perturbations for the found tokens; and 3) select the most influenced perturbations. Experimental results on three publicly released NMT systems show the attack approach can significantly decrease NMT efficiency.

**Summary Of The Review:**

The paper is very interesting and the contribution is clear, but the claims are not well supported by the existed experiments. Therefore, I give 5 (weak reject) at this moment and would like to increase/decrease my review score after the author response.

---

> ### Author Response · Authors · 2021-11-23
> **Uploaded Rebuttal**
>
> We thank the reviewer for the feedback on our work. The response to all the queries has been uploaded as an additional document. The General Response is for the queries asked by multiple reviewers. Following that, we address the queries of each specific reviewer.
>
> 1. Apply Output Length as the Evaluation Metric: The results are presented in the general response link.
>
> 2. Configuration of the Maximum Length: We follow the default maximum length of the models downloaded from the websites. For other maximum length settings, we add another experiment and the results are shown in the general response link.
>
> 3. Attacking online NMT models: Considering the legal and ethical factors, we did not conduct online experiments. However, the evaluation in this paper is conducted on the NMT models downloaded from HuggingFace (https://huggingface.co/).
> The victim models are the back-end models that are used for online translation services. Each victim model corresponds to an online translation service on the HuggingFace website~(we list the URL of the translation service in Table 2 of the paper).
> We download the NMT models from the websites to local machines directly, then conduct the experiments.  Theoretically, our experiment results indicate that the generated efficiency adversarial examples can also slow down the HuggingFace online translation service.  We provide some generated adversarial examples that can be used to test the online NMT API https://huggingface.co/Helsinki-NLP/opus-mt-en-de.

---

### Official Review · Reviewer_mDAN · 2021-11-03

**Correctness:** 3
**Technical Novelty And Significance:** 3
**Empirical Novelty And Significance:** 3
**Recommendation:** 3
**Confidence:** 4

**Main Review:**

Strength:
1. The paper is well-written.
2. This paper study a new research topic for attacking neural models.

Weakness:
1. Efficiency attack is a new research topic and thus this paper seems novel. However, for NMT efficiency attack is not as valuable as accuracy attack. As mentioned in the paper, efficiency is dependent on the length of the output sentence, which is controlled not only by the NMT model itself but also by some heuristics (such as length penalty and max-length). More importantly, it is easy to defend the efficiency attack by using some other heuristics. For instance, one can simply use the length ratio between source and target sentences to constrain the beam search algorithm. This paper does not take into account these decoding heuristics. Therefore, the overall contribution of the current version is limited.

2. The efficiency is closely dependent on the length of the output sentence, but the experiments do not report the efficiency about the length. In addition, the paper does not mention details about the beam search algorithm for decoding such as max-length.


**Summary Of The Paper:**

This paper considers to attack an NMT system from the perspective of efficiency. At a high level, it aims to slightly modify the input such that the target NMT system outputs a long translation, leading to decreased decoding efficiency. To this end, it proposes a method to guide the input modification such that the probability of EOS is as low as possible. Experiments show that the proposed method outperforms the baseline methods in terms of efficiency attack.


**Summary Of The Review:**

This paper study a new research topic for attacking neural models, i.e., efficiency attack, but it is not as meaningful as accuracy attack.

---

> ### Author Response · Authors · 2021-11-23
> **Uploaded Rebuttal**
>
> We thank the reviewer for the feedback on our work. The response to all the queries has been uploaded as an additional document. The General Response is for the queries asked by multiple reviewers. Following that, we address the queries of each specific reviewer.
>
>
> 1. Meaningful of the Proposed Attack:
> Our work shares the motivation of accuracy-based adversarial attacks, and has a unique real-world impact, because of the following reasons: (1). Neural Machine Translation models are more commonly used than traditional machine translation models because NMT models can capture \textbf{long dependencies} in sentences. However, the ability to handle long dependencies brings in a new risk, dead loops. There are two mechanisms to avoid dead loops in NMT models, \ie  (i) set a constant maximum length, (ii) set the maximum length according to the input length. However, the effectiveness of these two mechanisms against
> In this paper, we apply the first mechanism and set the maximum length according to the default configuration files. We evaluate the effectiveness of efficiency adversarial examples under the second maximum length setting and show the results in Table 1 of the general response. (2). One of the main goals of investigating vulnerabilities is to raise community concerns, and the fixes for the vulnerabilities are usually straightforward once the vulnerabilities are exposed, \eg preventing buffer overflow simply requires checking memory boundary when writing unsafe memory. In the machine learning community, accuracy-based adversarial attacks have already raised the concern of the committee and the committee is working on defense against accuracy-based attacks. However, efficiency-based adversarial attacks are currently ignored by the research academia and industry at the current stage. We study the online translation service of HuggingFace (https://huggingface.co/), which is a commercial company that provides the online NLP model API. We randomly select 100 back-end  NMT models from HuggingFace's API services (https://huggingface.co/models?pipeline_tag=translation&sort=downloads) and parse each NMT model's configuration file to check how they set the maximum length. Unfortunately, the selected models all set a constant maximum length, and the maximum length is larger than the maximum length in our evaluation~(\ie, the maximum length range from 200 to 1024 with the average maximum length of 501).
> In this paper, we first characterize these new vulnerabilities and want to raise the concern of the committee.
>
>
> 2. Configuration of the Maximum Length in our Evaluation: We present the detail in the general response link.

---

### Author Response · Authors · 2021-11-23
**General Response**

The general response to reviewers' comments.
We add some experiments to address the reviewers' concerns, the details are put in the following link.
https://openreview.net/attachment?id=zfmB5vgfaCt&name=supplementary_material

---

### Decision · Program_Chairs · 2022-01-20

**Decision:**

Reject

**Comment:**

The papers studies a novel problem and proposes an interesting algorithm. That said, the reviewers question the motivation of the paper. That is whether this method presents a viable attack on existing MT systems. The attack is not black box and MT systems often have an output length threshold beyond which the output is trimmed. Given the motivational concerns, I recommend that the paper is revised and resubmitted to other venues.